# Classification of Tree Species in the Process of Timber-Harvesting Operations Using Machine-Learning Methods

Fedor Svoikin [1], Kirill Zhuk [1], Vladimir Svoikin [2], Sergey Ugryumov [1], Ivan Bacherikov [1], Daniela Veas Iniesta [3] and Anatoly Ryapukhin [3,*]

[1] Saint Petersburg State Forest Technical University, Institutskiy Lane 5, 194021 Saint Petersburg, Russia
[2] Syktyvkar Forest Institute (Branch), Saint-Petersburg State Forest Technical University Named after S.M. Kirov, Lenin Street 39, 167982 Syktyvkar, Russia
[3] Moscow Aviation Institute, Volokolamskoe Highway 4, 125993 Moscow, Russia
[*] Correspondence: ryapukhin.a.v@mail.ru

**Abstract:** This article presents the constraining factors that limit the increase in the efficiency of logging production by modern multi-operation machines operating on the Scandinavian cut-to-length technology in the felling phase, namely the selection and registration of wood species. The factors for creating a complete architecture of a fully connected neural network (NN) are given. The dependence of the prediction accuracy of a fully connected NN on a test sample on the size of the training dataset, and an image of the dependence of the prediction accuracy on the number of trees in the random forest method for image classification is shown. For a fully connected NN, a sufficient number of images and a test sample size were established for training, using tree-trunk breed-class labels as target values. A selected list of trees was given, with the size of the training sample of images presenting a problem for the classification of tree trunks using the random forest method. The aim was the discovery of the optimal number of trees necessary to achieve prediction accuracy.

**Keywords:** forestry machines; harvester; harvester operator; logging operations; stem species; image classification; neural networks; random forest; cut-to-length technology

## 1. Introduction

In the timber industry of the Russian Federation, at this stage of the development of the production process, the primary phase of logging is an important stage, in which modern multi-operation forest machines are used, working mainly on Scandinavian cut-to-length technology, each unit of which is controlled by a logging equipment operator. Numerous sensors, powerful equipment and smart technologies allow forest machine operators to harvest, on average, a range of 150 to 240 m$^3$ of wood (an acceptable volume of wood harvesting) in one full work shift in Russia [1–4].

In the current dynamically changing economic conditions, the issue of justifying the choice and operation of new logging equipment becomes the most relevant. The forestry equipment market in the Russian Federation is almost 100% dependent on imports (John Deere, Ponsse, Komatsu, Rottne, Volvo, Logset and CAT). Traditional imported solutions for cut-to-length Scandinavian logging technology are becoming unavailable for the Russian market for several reasons. Therefore, loggers in the Russian Federation are currently considering the refurbishment of existing equipment and the purchase of new Russian or foreign equipment available on the market (manufactured most often by the People's Republic of China, namely in 2023 by SANY, SDLG and LIU GONG).

Among the Russian solutions for Scandinavian cut-to-length technology, it is worth noting the future products of the KAMAZ: KAMAZ-1010 felling–lopping–buckering machine named Harvester (VSRM) and the KAMAZ-1030 wheeled log picker named Forwarder (KS).

The VSRM KAMAZ-1010 is a four-wheel-drive articulated vehicle with an 8 × 8 wheel arrangement and a curb weight of 22,000 kg. The KAMAZ-1010 is equipped with a 325-horsepower Euro-3 diesel engine and a two-stage transfer case NAF VG75 (manufactured in Germany). Its overall dimensions are length 11.5 m, width 3 m, and height 3.99 m. The KESLA proLOG and xLogger systems (manufactured in Finland) can be used as measurement and control systems. However, KAMAZ is currently developing a Russian harvester head (HG) manufactured by Uralvagonzavod in Nizhny Tagil, which will appear in 2023.

The KS KAMAZ-1030 and VSRM KAMAZ-1010 have articulated designs, and have a wheel arrangement of 8 × 8. Most likely, the KS KAMAZ-1030 is equipped with the same engine. The hydraulic manipulator is identical to Kesla (manufactured in Finland).

At the moment, the only competitor in the sector of the Scandinavian cut-to-length technology of KAMAZ products is AMKODOR (manufactured in the Republic of Belarus).

In the product line of AMKODOR, there are analogues of VSRM KAMAZ-1010 and KS KAMAZ-1030. The VSRM "Amkodor FH3081" and KS "Amkodor FF1681" of a heavy class with a wheel arrangement of 8 × 8 were presented in 2020. However, AMKODOR has been developing competencies in the production of logging equipment for 18 years, while KAMAZ has only been developing for 2 years.

It should be noted that in 2002 (20 years ago) KAMAZ PJSC tried to enter the tractor equipment market. In 2002, the KT-240 wheeled tractor was created at a plant on the basis of components and assemblies of the KAMAZ vehicle. It was almost completely assembled from components of its own production; however, the cab was taken from a VT-130 tractor (Volgograd). A 240-horsepower KAMAZ V8 diesel engine was used as a power unit. The speed range is from 2.3 to 40 km/h. The KT-240K could perform all agricultural work: plowing, cultivating land, fertilizing and harvesting. An experimental batch of 12 machines was produced. After testing, it became clear that the model turned out to be quite successful. However, all the pluses were outweighed by one big minus: the price was too high. Therefore, this tractor did not go into mass production.

In 2007, KAMAZ tried to start a joint production of tractors with the Italian holding ARGO SPA, which at that time owned the Landini (Italy) and McCormick (UK) factories. The choice was stopped at McCormick (UK) according to the "balance of price–quality" criteria. The McCormick plant was then closed and it became possible to bring equipment and equipment from there and quickly arrange the assembly of tractors in Naberezhnye Chelny. They were going to produce the KAMAZ T-215 model with an operating weight of 7205 kg. The tractor was equipped with a 5.9-L turbodiesel engine, whose maximum power with the power management system reached 229 hp, and a smart electronic transmission. The driver's cab had the highest level of comfort at that time.

Plans were grandiose: 2500 tractors at the first stage in 2008; 4000 tractors in the second phase in 2009; 8000 in five years. The gradual localization of the model were envisaged, with the transition to the Cummins engines manufactured by CUMMINS-KAMA JV, and the release of T-185 and T-200 modifications of lower power. The key sales markets were Russia, Kazakhstan and Turkmenistan.

In 2009, KAMAZ delved deeper into agricultural topics: two more joint ventures were created with Case New Holland (CNH), which was part of the FIAT group. The first was for the production of tractors, other agricultural machinery and construction equipment (CNH-KAMAZ Industry/CNH-KAMAZ Industrial). The second was for their sale and service (CNH-KAMAZ Commerce/CNH-KAMAZ Commercial). A subsidiary enterprise LLC "Kamsky Tractor Plant" (KamTZ) was created. In 2010, the assembly of new models of tractors T8050, T9040, T9060 with a power of 325 to 535 hp, as well as CSX7060 combined with a power of 272 hp and CSX7080 with 300 hp began in Naberezhnye Chelny. In addition, the production of tractors XTX-185 and XTX-215 continued.

In April 2013, CNH-KAMAZ Industry JV produced its 1000th tractor. At the same time, plans were announced to expand the model range, switch to the small-scale assembly of tractors in the near future and achieve a 45% level of localization. In addition, in November,

KAMAZ unexpectedly announced its intention to withdraw from CNH-KAMAZ Industry JV and sell its share to the Italians. Representatives of KAMAZ then said that such a decision was part of the strategy to remove non-core assets, which, as part of the efficiency-improvement program, was adopted at the end of 2011. Experts gave the following explanation: although the joint venture worked with a profit, it required large investment for further development, and KAMAZ could not afford it.

Given the negative experience in the agricultural industry, as well as the trend towards SKD assembly using a large number of key foreign components, at the moment, the development of Russian solutions by KAMAZ for the logging industry is at the R&D stage. It should be noted that design solutions are carried out using the method of reverse engineering foreign analogues.

AMKODOR is developing in a similar way.

Presently, in the Russian Federation there are serial technical solutions that are already used in logging, e.g., KS TROM 20 based on a snow and swamp vehicle with ultra-low pressure pneumatics. This type, based on all-terrain vehicles, has successfully established itself in the oil and gas industry of the Russian Federation over the past 10 years.

Taking into account the trends in the development of forestry engineering, special attention for monitoring the wood supply chain should in any case be given to control-measurement systems of a VSRM harvester head.

In conditions of limited or no access to the current repair and maintenance of modern logging equipment, especially control and management systems (CAN architecture according to the StanForD 2010 protocol), the main problem in the logging process is insufficiently qualified specialists who do not have the proper competencies or access to them, in order to undertake the required adjustment of the forest machine so as to ensure the continuous operation of the base machine for control and management systems. To overcome the competency gap, novice operators (including operators with little experience) must seek the help of experienced operators. Based on the analysis of the requests of logging companies and the labor market, an experienced and qualified operator in the Russian Federation is a specialist who has continuous experience working on a forest machine for more than 5 years. It should be noted that, in reality, there are catastrophically few such specialists, and therefore, if difficulties arise for novice specialists, time delays and downtime are formed in the logging process chain. Even operators with sufficient work experience often do not have the necessary competencies due to the maintenance of the equipment they work on under the "full-service" program. The lack of competencies of experienced operators with "full service" appears due to the transfer of powers for servicing equipment (up to replacing high-pressure hoses and sharpening saw chains) from the operator to the service mechanic of the equipment supplier. According to a survey of focus groups of instructors and service mechanics of modern imported logging equipment in the Russian Federation, the operator uses only 10% of the digital capabilities of harvesters due to traditional non-digitalization and non-digitization of the industry, so the task of determining the species of tree trunk from an image based on the methods of a fully connected NN and random forest to improve the efficiency and productivity of the overall felling–branching–bucking phase is relevant. Nevertheless, at the moment, in the Russian Federation, there is a lack of research on this topic.

## 2. Theoretical Basis

One of the challenges for operators is setting up the soft buttons on the machine control joysticks to select the tree species of the tree trunk to be felled. Every time the operator changes from one machine to another, he needs to program the breed-selection buttons to suit his personal ergonomics. Each breed has its own separate button. For example, when working in a mixed forest (birch, spruce, pine), it is necessary to program three separate buttons. Considering the work in the system settings, this can take a considerable amount of time. Additionally, when processing a tree trunk, immediately after the capture and before the felling process, the operator must press the button for the species of the harvested

trunk to register the assortments that will be obtained during the processing of the trunk. For inexperienced operators, this process can take 2–3 s [5,6]. With the number of harvested trunks being 500–1000 pieces per work shift, there is a downtime of 1000–2100 s. When working in two shifts of 10 h, a delay of 2000–4200 s is formed on average per day. Thus, the task of automatically determining the species of a tree trunk to increase the processing speed and the overall productivity of the harvesting cycle is an urgent task.

To create a complete architecture of a fully connected NN, it is necessary to initially calculate the sizes of the input and output layers. The input layer is a vector of image pixel values. The data for training machine-learning methods will be the sets of images obtained from the camera, which is installed in the operator's cabin of the FDBM (felling–delimbing–bucking machine) [7,8].

A feature of a fully connected NN is that when working with images, both in grayscale and color ones, a vector of numerical values must be fed to the input layer. The images are presented as matrices. Under current conditions, images are chromatic and are represented as a set of matrices, where each matrix corresponds to one channel (red, green, blue).

The aim of the work is to study classifiers for the problem of determining the species of a tree trunk from an image based on the methods of a fully connected NN and random forest.

### 3. Methodology

The Python programming language was used for the research calculations. Model NN was built using the keras framework in Python. PyCharm was used as the integrated development environment (IDE) for Python research. Machine-learning libraries sklearn and tensorflow were applied for models to predict stem species by images. The NumPy library was used for metrics calculations. The Seaborn library was used for result visualization.

When determining the size of the input layer, it is necessary to extrude the image, consisting of matrices, into one large vector with pixel values. Image sizes were selected from the following list:

- $64 \times 64$ pixels
- $128 \times 128$ pixels
- $256 \times 256$ pixels
- $512 \times 512$ pixels

The image sizes were chosen empirically. Converting a $64 \times 64$-pixel image will result in an input vector of size 12,288 values, since we are working with color images, therefore, $64 \times 64 \times 3$ is needed for the calculation. For a $128 \times 128$ image, we acquire 49,152 values. For $256 \times 256$, the input vector will have a length of 196,608 values. For $512 \times 512$, there will be 786,432 values. The optimal value for image sizes for the input layer is $128 \times 128$, since it is at these parameters that the images contain enough information to be used in the classification problem. Images with sizes of 256 and 512 contain even more information, but at these sizes, the input layer will have over 1 billion trainable parameters, which will lead to extremely slow training of a fully connected NN when using an Nvidia GeForceTM RTX 3050 graphics card. When using a $64 \times 64$ image, there may not be enough information.

Thus, by empirically comparing several image sizes to form an input vector, a size of $128 \times 128$ was chosen.

When determining the size of the output layer, it is necessary to take into account the number of stem species that will be classified. In this case, a fully connected NN is built to classify three types of stem species:

- birch
- spruce
- pine

The output vector will have, in this case, three values. For an image with a birch trunk, the vector will be [1, 0, 0], for a spruce [0, 1, 0] and for a pine [0, 0, 1]. Thus, the output layer will have three values.

The input layer will have a size of 49,152, which means that when using sufficiently large internal hidden layers of a fully connected NN, it will be difficult to train them. Hence, the optimal size of the second layer in the NN will be 2048 neurons. Thus, the number of trainable parameters between the first and second layers will be 49,152 × 2048, which is equal to 100,665,344. This is enough to optimally train the parameters between layers. The third layer has a size of 1024 neurons, and the number of trainable parameters is 2,098,176. For the fourth layer, the size is 512 neurons and 524,800 trainable parameters. The fifth layer has 32,832 trainable parameters and a size of 64 neurons. The output layer has three neurons and 195 trainable parameters.

Thus, it turns out that the total number of trainable parameters in the NN is 103,321,347. Each value is represented by 4 bytes; therefore, the hard disk space occupied only for trainable parameters will be 387 megabytes. The NN has sigmoid as activation function at each of the four hidden layers. The last layer uses a SoftMax function. The cross-entropy function was used as a loss function.

## 4. Results

After creating the architecture of a fully connected NN, it is necessary to collect and process a common set of images, which will contain 8000 images of each breed. In total, the dataset will contain 24,000 color images. Figure 1 shows an example of a training picture, and Figure 2 shows the markup of this trunk. A mask with markings is superimposed on the image of the trunk, which allows the removal of unnecessary noise that adversely affects the quality of classification when using a fully connected NN.

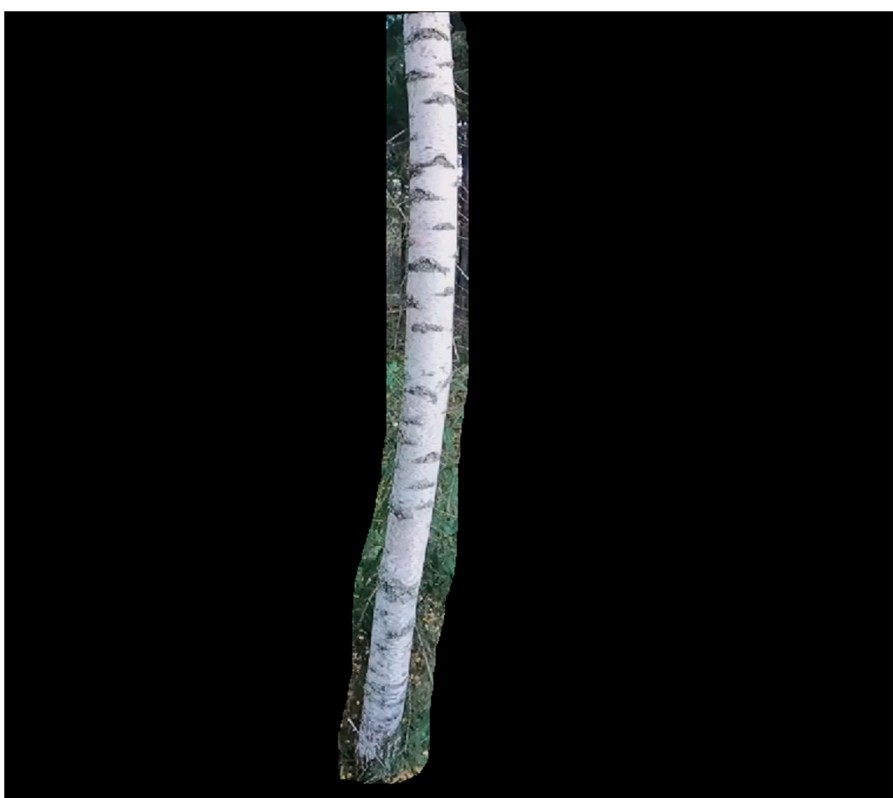

**Figure 1.** An example of an image from the general sample with a birch trunk.

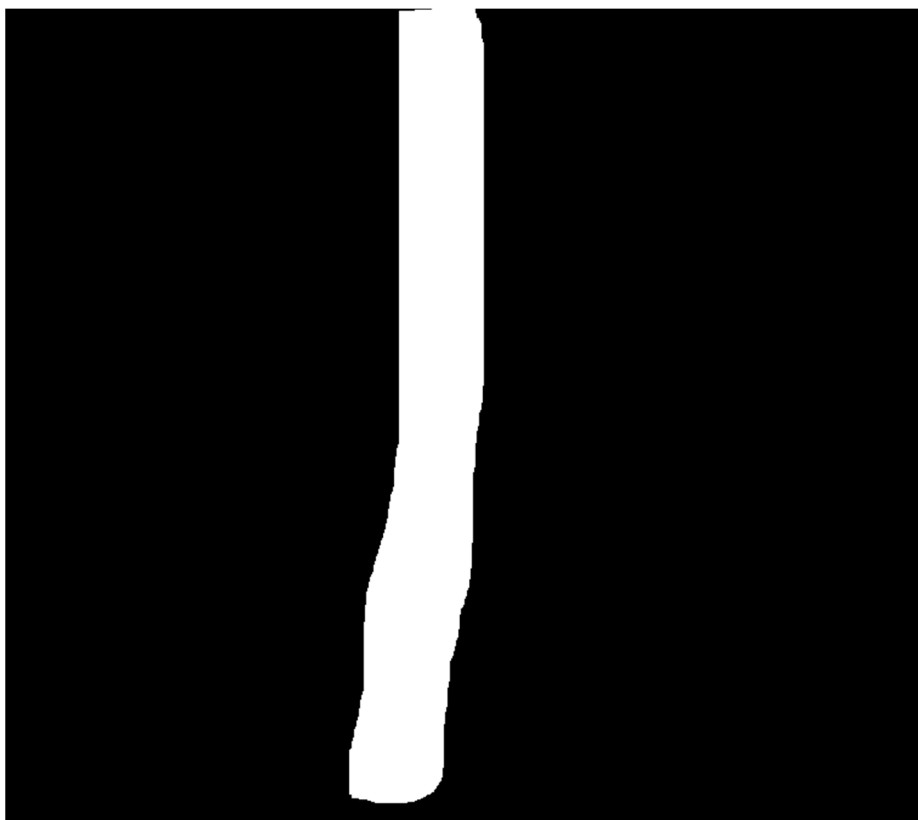

**Figure 2.** An example of marking up an image with a birch trunk.

The preparation time required for 24,000 images and their markup is about 200 person-hours ((24,000 (img) × 30 (s/img))/3600 (s)). When working every day for 1.5–2 h, this dataset is created in 100–110 days. Estimated preparation time for the overall dataset was based on the average build and markup time for the image. The average time for one image is 30 s.

When training and testing a fully connected NN, it is necessary to take into account the fact that the data must be divided into two groups. The first group will include images for the training set, and the second group will include images for the test set. Groups should not share images. Before training and testing for each image, a vector of three values is compiled, in which a unit is put down in accordance with the breed.

Thus, pairs of elements are obtained—an image and a vector of stem species. The received data are converted into arrays using the NumPy library of the Python programming language [9–12]. The determination of the species composition of plantations by the image of trunks with subsequent data augmentation was presented [13], but at the moment there is a lack of research on this topic. At the moment, the current results of the work on the project have been published [14–19] and discussed (VI and VII All-Russian Scientific and Technical Conference "Forests of Russia: Politics in St. Petersburg Forest Technical University, Industry, Science, Education", scientific and practical conference "February Readings" based on the results of the Syktyvkar Forest Institute research work in 2021 and 2022) and also supported by a grant from the Council for Grants of the President of the Russian Federation for state support of young Russian scientists—candidates of science (Competition—MK-2022), number MK-2025.2022.4. The size of the training sample is 75% of the total set, and the test sample is 25%. To determine the optimal sample size, the dependence of the forecasting accuracy on the sample size is set. To do this, the creation of several datasets is required:

- 225 training and 52 test pairs
- 1125 training and 262 test pairs

- 2250 training and 525 test pairs
- 4500 training and 1050 test pairs
- 11,250 training and 2625 test pairs
- 18,000 training and 4200 test pairs

Figure 3 shows an image of the constructed dependence of the prediction accuracy of a fully connected NN on a test set on the size of the training dataset.

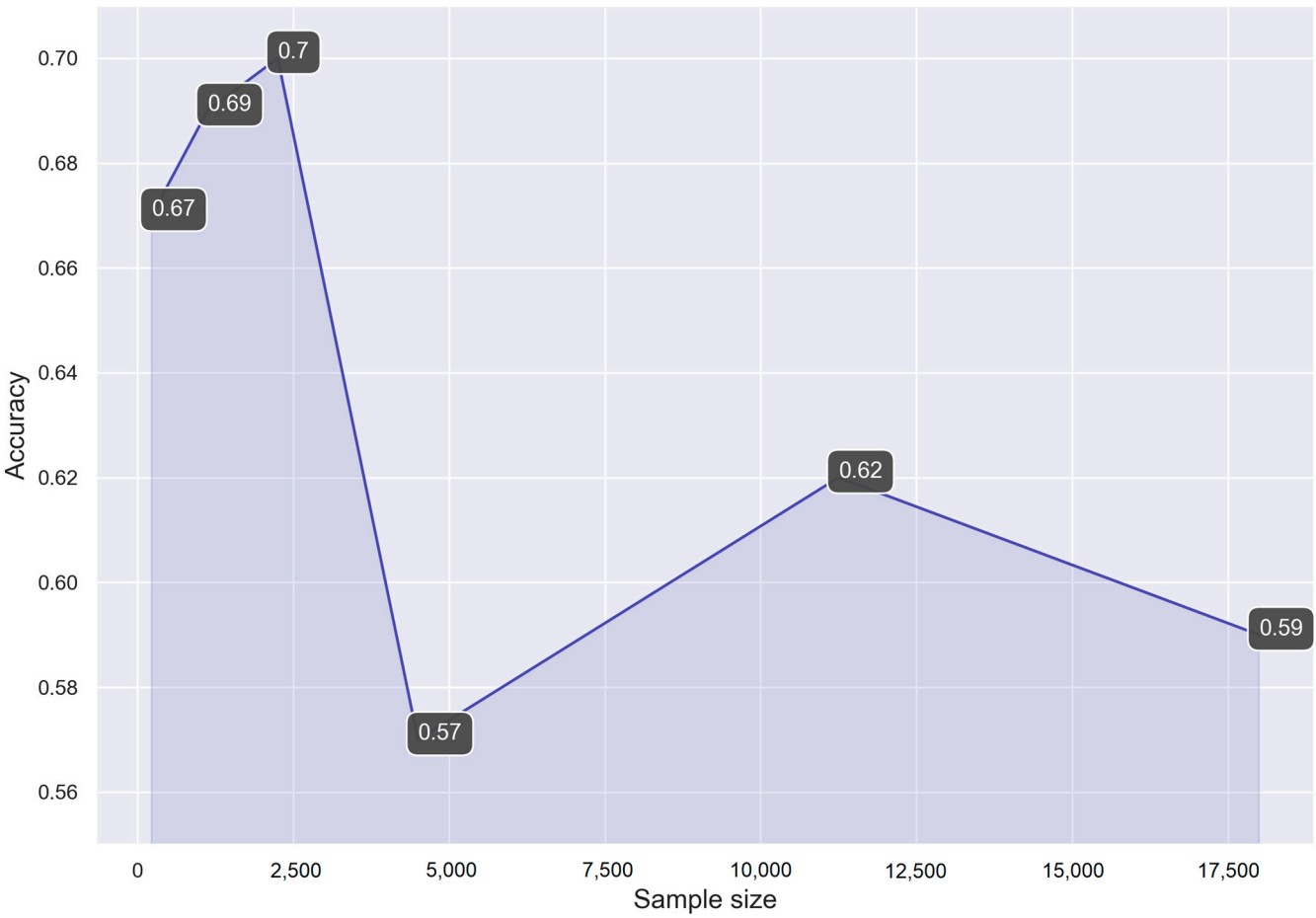

**Figure 3.** The dependence of forecasting accuracy on sample size.

With a training sample size of 2250 images, a prediction accuracy of 0.7 is obtained; however, since the shape of the trunks is quite variable during the operation of the FDBM, it is necessary to increase the sample size. The optimal sample size for classifier training is 11,250 images. With such a sample size, a prediction accuracy of 0.62 is achieved. This value is less than with 2250 images, but contains five times more different data, including different shapes of tree trunks.

The NN is not the only method used for image classification. One of the most relevant methods is random forest. The random forest method uses the bootstrap approach, in which the final classification decision is formed based on most votes. A set of decision trees is created in a random forest. Each tree is fed, in order, a training pair of values. The final result of the classification is based on the calculation of the predicted values of each tree.

The training sample for the random forest method has different target values. The fully connected NN had output vectors of size 3. In this approach, you need to use a single numeric value that will correspond to the class of the tree trunk. In the current study, the class numbers are distributed as follows:

- birch 0
- spruce 1

- pine 2

Thus, each image will have a corresponding numeric value of the class label. The resulting image–label pairs are fed to the random forest classifier.

With a training sample size of 11,250 128 × 128 images and 1000 training trees, the total training time of the algorithm will be 568 s. With the same training sample and the number of trees equal to 100, the training time will be 60 s. It is important to determine the partition quality criterion in the process of learning the decision tree. Empirically, the gini method was chosen.

The entropy method takes longer because a logarithm needs to be calculated. When conducting research into a fully connected NN, it was found that using 11,250 images for training is sufficient. This approach also uses a test sample of 3750 images. The target values are the class labels of the tree-trunk species. The list of the number of trees for the study was chosen to be 50, 100, 200, 300, 500, and 1000. Figure 4 shows the dependence of the prediction accuracy on the number of trees.

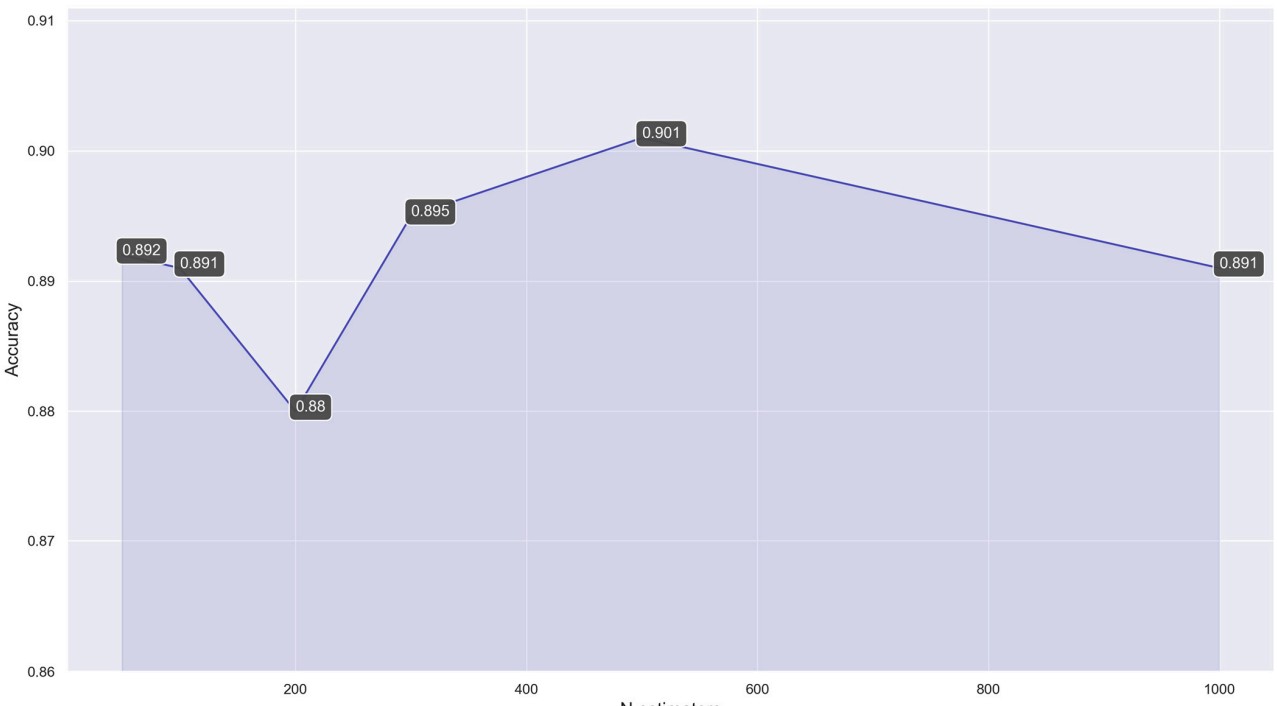

**Figure 4.** The dependence of forecasting accuracy on the number of decision trees.

As a result of the research, we present the proposed method in the form of an algorithm/flowchart (Figure 5).

To train the segmentation NN, it is necessary to designate the function by which the error will be calculated for each training example. In this case, we will use the Sørensen–Dice coefficient, because the initial image will be compared with the result, which NN predicts. For the test sample, the average value of Sørensen–Dice coefficient was 0.58 for 80 epochs of segmented NN training. The comparison of the average value of the Sørensen–Dice coefficient from the number of epochs is shown in Table 1.

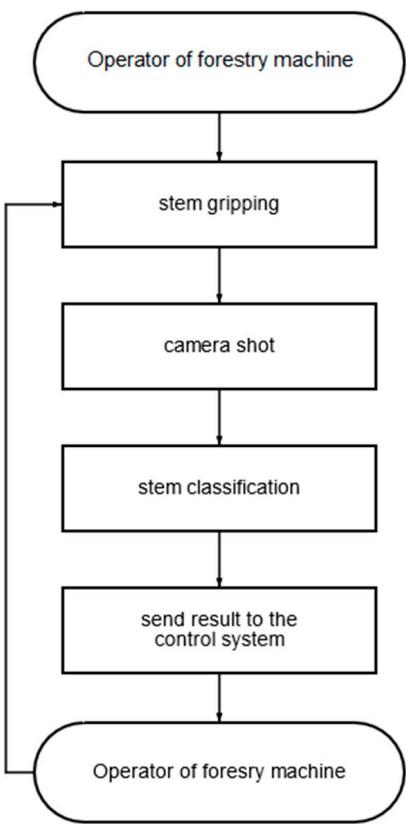

**Figure 5.** Research result in the form of an algorithm/flowchart.

**Table 1.** Magnitude of the average value of Sørensen–Dice coefficient depending on the number of epochs of training.

| Number of Eras | Dice |
| --- | --- |
| 10 | 0.43 |
| 20 | 0.37 |
| 30 | 0.47 |
| 50 | 0.54 |
| 70 | 0.53 |
| 80 | 0.58 |
| 100 | 0.52 |

It is worth recalling that training was conducted using stochastic gradient descent and involved updating the parameters of the segmentation NN after each passage of one picture from the training dataset.

A comparison of Sørensen–Dice coefficient values when dividing the raw data into training and test samples in the ratio 3:1 and 4:1 is shown in Table 2.

**Table 2.** Comparison of the average Sørensen–Dice coefficient scores for different partitions of the original data.

| Number of Eras | Dice (0.75) | Dice (0.8) |
| --- | --- | --- |
| 10 | 0.43 | 0.34 |
| 20 | 0.37 | 0.36 |
| 30 | 0.47 | 0.41 |
| 50 | 0.54 | 0.43 |
| 70 | 0.53 | 0.46 |
| 80 | 0.58 | 0.45 |
| 100 | 0.52 | 0.41 |

## 5. Discussion

The value of the Sørensen–Dice coefficient value for perfect segmentation will be as high as one. A coefficient value between 0.9 and 1 would be considered an excellent result, a good result between 0.7 and 0.9, and a satisfactory one between 0.5 and 0.7. Satisfactory results were obtained for the number of epochs equal to 50, 70 and 80. Such indicators are most likely to be related to the markup of the masks for the original images. All 100 images, which are involved in the training of the segmentation NN, were collected from the Internet, and the markup of masks was completed manually using a specialized graphics editor. In some cases, artifacts in the form of separate areas of $1 \times 1$ pixels could occur during the markup, which would affect the quality of NN training. However, the pictures from the initial dataset were not completely marked up, i.e., a mask was not created for all of the tree trunks that were depicted on them, but only for those that stood out the most. In addition, it turned out that the segmentation NN resulted in image masks, on which were found parts of the trunks, which were not marked in the dataset, and, thus, when calculating the Sørensen–Dice coefficient, the values could be different, because there would be more areas that overlapped. Therefore, in this case, the resulting value of 0.58 for 80 training epochs can be considered good. Dividing the original dataset into training and test samples in the ratio 4:1 showed a worse result than 3:1 split. One would assume that by increasing the training sample size, the result of the segmented NN should be better, but because we increased the training sample, the test portion decreased. The original 100 images contain 20 images taken from the outside of the harvester. Perhaps we should replace these 20 images with a view from the cabin of the forest machine in which the operator sits.

It can be seen from the dependence that even with the number of decision trees equal to 50, an accuracy value of 0.892 is achieved. With this approach, the training time of the algorithm is 34 s. However, when using 500 decision trees, a prediction accuracy of up to 0.901 is achieved, which is about 1% more than with 50 trees. In this case, the time spent on training is 291 s. Thus, it was found that with a training sample size of 11,250 images with the problem of classifying tree trunks using the random forest method, the optimal number of trees is 500 with an achievable prediction accuracy of 0.901.

Automatic methods in the forestry industry are used to measure the total volume of logs in log trucks. A mobile phone with the application installed in it could use convolutional NN for tree-stem breed classification. One of the areas of interest in the forest industry is the assessment of the volume of forest reserves. Given such a task, unmanned aerial vehicles (UAV) with LiDARs installed on them are used. In the future, point clouds could be used to obtain segmented tree fragments using the a specialized neural network. In [13] the vgg19_bn pretrained model showed accuracy of 0.87 and used an un-trained model accuracy equal to 0.83. The classification of four tree species was considered—spruce, pine, birch and aspen. Thus, the random forest approach shows 0.9 accuracy score. That means the pretrained vgg19_bn model has less accuracy score by 0.03 and by 0.07 compared to the unpretrained model.

## 6. Conclusions

As a result of the research into fully connected NN, it was found that the use of 11,250 images for training is sufficient (a test sample of 3750 images) when using tree-trunk breed-class labels as target values. With the selected list of the number of trees for the study—50, 100, 200, 300, 500, 1000—and with a training sample size of 11,250 images given the task of classifying tree trunks using the random forest method, the optimal number of trees is 500 with an achievable prediction accuracy of 0.901.

This study was conducted to analyze the approaches that can be used for tree-trunk segmentation. Understanding the value of the timber volume in a harvested area before the process begins allows for advance planning of further actions in the harvesting chain, which leads to sustainable management. If a quick estimation of available resources is needed, and it is advisable to use UAV that are equipped with LiDAR. Information from these devices is processed using modern machine-learning methods and models such as

VoxNet. This approach identifies the tree crown and most of the trunk well; however, the part of the tree in the trunk is segmented residually poorly, and the average error, when calculating diameter at breast height (DBH) parameter, was 15.85%. Finding the trunk of a tree using segmentation NN on a picture that is obtained from the forest machine cab will be useful for the automatic classification of species in the bucking process. This approach will save time in setting up the forest machine for a particular operator, and since some operators may not have enough knowledge to program the names of species on the joystick buttons, the use of machine-learning methods to solve this problem will be optimal. The obtained results of the segmentation NN in this work can be used to calculate the volume of the specified tree before cutting, based on the calculated DBH parameter. This will allow a comparison of the volume of harvested wood raw material with the data, which is calculated by the onboard control system of fuzzy dynamic characteristic modeling (FDCM) with the help of sensors measuring the trunk length and diameter, and if there are significant deviations, it will be necessary to pass the process of calibration with a measuring fork on FDCM.

**Author Contributions:** Conceptualization, S.U., I.B., D.V.I. and A.R.; methodology, K.Z. and V.S.; software, S.U. and I.B.; validation, F.S., K.Z., V.S., S.U., I.B., D.V.I. and A.R.; formal analysis, F.S. and K.Z.; investigation, V.S. and A.R.; resources, S.U. and I.B.; data curation, D.V.I. and A.R.; writing—original draft preparation, F.S. and A.R.; writing—review and editing, V.S., S.U. and I.B.; visualization, K.Z.; supervision V.S. and S.U.; project administration, A.R.; funding acquisition, D.V.I. and A.R. All authors have read and agreed to the published version of the manuscript.

**Funding:** This research received no external funding.

**Data Availability Statement:** Not applicable.

**Conflicts of Interest:** The authors declare no conflict of interest.

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
