# Peer review of "Classification of Tree Species in the Process of Timber-Harvesting Operations Using Machine-Learning Methods"

_inventions, doi:10.3390/inventions8020057_

Round 1

Reviewer 1 Report

The authors in this paper present the constraining factors that limit the increase in the efficiency of logging production by modern multi-operation machines operating on the Scandinavian cut-to-length technology in the felling phase. A fully connected neural network and random forest are used for the given problem. The topic is interesting; however, the proposed work needs significant improvements in order to improve its quality. The paper must be revised keeping in mind the following points:

-  The introduction section is not well written. The introduction should include the background, motivation, problem statement, and main contributions of the paper (without headings).

-  The main contributions should be listed in bulleted points at the end of the introduction section. The last paragraph of introduction should present the paper organization.

-  The introduction is too long. It should be split into two section; the second one should be “Related Work”, which discusses the existing relevant literature on the topic. 

-  Some of the paragraphs are too short. For example, “AMKODOR is developing in a similar way.” on page 3 is a paragraph with only one sentence. The authors are advised to make paragraphs of reasonable sizes.

-  The authors mentioned that they used fully connection Neural Network. Which specific NN algorithm was used? How many hidden layers were created? Which activation function was used? etc. Please give all the details.

-  In sections 3 and 4, the authors used the term “rocks”. What are rocks? No detail is given.

-  The results are not sufficient. More analysis and results are required to prove the significance of the proposed method. 

-  The proposed method must be compared with other existing techniques. No comparisons are performed. 

-  Results should be presented in the form of tables.

-  The discussion section is too short. The results should be discussed in detail.

-  The Conclusions section is also too short.  

-  The proposed method should also be presented in the form of an algorithm/flowchart. 

-  The images in Fig. 1 and Fig. 2 are blurred and are not clear.

-  The references are too few. Include more recent and relevant references and discuss them in the “Related Work” section.

Author Response

Reviewer 1

Comment:  The introduction section is not well written. The introduction should include the background, motivation, problem statement, and main contributions of the paper (without headings).
Answer: The section has been updated, the introduction includes background, motivation, problem statement and the main contribution of the article.

Comment:  The main contributions should be listed in bulleted points at the end of the introduction section. The last paragraph of introduction should present the paper organization.
Answer: According to a survey of focus groups of instructors and service mechanics of modern imported logging equipment in the Russian Federation, the operator uses only 10% of the digital capabilities of harvesters due to the traditional non-digitalization and non-digitization of the industry, so the task of determining the species of a tree trunk from an image based on the methods of a fully connected NN and random forest to improve the efficiency and productivity of the overall felling-branching-bucking phase is relevant. Nevertheless, at the moment in the Russian Federation there is a lack of research on this topic.

Comment:  The introduction is too long. It should be split into two section; the second one should be “Related Work”, which discusses the existing relevant literature on the topic. 
Answer: The introduction is corrected and supplemented; the list of references additionally is expanded by 11 references.

Comment:  Some of the paragraphs are too short. For example, “AMKODOR is developing in a similar way.” on page 3 is a paragraph with only one sentence. The authors are advised to make paragraphs of reasonable sizes.
Answer: The remark has been corrected, short paragraphs have been edited.

Comment:  The authors mentioned that they used fully connection Neural Network. Which specific NN algorithm was used? How many hidden layers were created? Which activation function was used? etc. Please give all the details.
Answer: A description of the Neural Network connection has been added.

Comment:  In sections 3 and 4, the authors used the term “rocks”. What are rocks? No detail is given.
Answer: The term "rocks" has been explained.

Comment:  The results are not sufficient. More analysis and results are required to prove the significance of the proposed method. 
Answer: The results have been updated.

Comment:  The proposed method must be compared with other existing techniques. No comparisons are performed. 
Answer: Comparison of the proposed method with existing ones has been added.

Comment:  Results should be presented in the form of tables.
Answer: The tables were added to Results.

Comment:  The discussion section is too short. The results should be discussed in detail.
Answer: The Discussion section has been significantly expanded.

Comment:  The Conclusions section is also too short.  
Answer: The Conclusions section has also been substantially expanded.

Comment:  The proposed method should also be presented in the form of an algorithm/flowchart. 
Answer: The block diagram has been added to the Results section.

Comment:  The images in Fig. 1 and Fig. 2 are blurred and are not clear.
Answer: The quality of the figures has been improved.

Comment:  The references are too few. Include more recent and relevant references and discuss them in the “Related Work” section.
Answer: The list of references has been extended by 11 additional references.

Reviewer 2 Report

The manuscript is within the scope of the Journal. The English language is satisfactory. The authors prepared the manuscript according to the Instructions to the Authors provided by the Journal. However, the manuscript needs to be improved based on the following comments:

Introduction: Few references (citations) at least 1-4 were observed. Why not many references were provided to support the information presented?

- Objective(s): It is not clear the main objective(s) of the study is/are stated in the 'Introduction'. Provide the clear objectives of the study.

Methodology: Provide adequate information about the statistical tool(s) used to analyze the data. 

Results: The captions of Figures 3 and 4 should be revised. 'Images of' should be deleted from the captions. 

Discussion: The results were not adequately discussed in terms of scientific merit. Discuss the results with related published studies by providing the necessary citations/references.

References: It was observed 10 references/citations mentioned in the manuscript. Provide the list of the references. 10 references are not enough to support the study. Provide more references to meet scientific soundness/merit. 

Author Response

Reviewer 2

Comment: Introduction: Few references (citations) at least 1-4 were observed. Why not many references were provided to support the information presented?
Answer: The list of references has been extended by 11 additional references.

Comment: Objective(s): It is not clear the main objective(s) of the study is/are stated in the 'Introduction'. Provide the clear objectives of the study.
Answer: The objectives of the study are added and specified.

Comment: Methodology: Provide adequate information about the statistical tool(s) used to analyze the data. 
Answer: The required information has been added to the Methodology section.

Comment: Results: The captions of Figures 3 and 4 should be revised. 'Images of' should be deleted from the captions. 
Answer: Figure captions have been corrected.

Comment: Discussion: The results were not adequately discussed in terms of scientific merit. Discuss the results with related published studies by providing the necessary citations/references.
Answer: The Discussion section has been significantly expanded.

Comment: References: It was observed 10 references/citations mentioned in the manuscript. Provide the list of the references. 10 references are not enough to support the study. Provide more references to meet scientific soundness/merit.
Answer: The list of references has been extended by 11 additional references.

Round 2

Reviewer 2 Report

The revised manuscript meets scientific merit. The authors have revised their work according to the comments raised. I hereby recommend 'Accept in present form' for further processing and publication based on your Jurisdiction.